



# Skill and independence weighting for multi-model assessments

Benjamin M. Sanderson[*1], Michael Wehner[†2], and Reto Knutti[‡3,1]

[1]National Center for Atmospheric Research, Boulder CO, USA
[2]Lawrence Berkeley National Laboratory, CA, USA
[3]ETH Zurich, Switzerland

November 2016

## 1  Abstract

We present a weighting strategy for use with the CMIP5 multi-model archive in the 4th National Climate Assessment which considers both skill in the climatological performance of models over North America as well as the inter-dependency of models arising from common parameterizations or tuning practises. The method exploits information relating to the climatological mean state of a number of projection-relevant variables as well as metrics representing long term statistics of weather extremes. The weights, once computed can be used to simply compute weighted means and significance information from an ensemble containing multiple initial condition members from co-dependent models of varying skill. Two parameters in the algorithm determine the degree to which model climatological skill and model uniqueness are rewarded; these parameters are explored and final values are defended with respect to the Assessment. The influence of model weighting on projected temperature and precipitation changes is found to be moderate, partly due to a compensating effect between model skill and uniqueness. However, more agressive skill weighting and weighting by targeted metrics is found to have a more significant effect on inferred ensemble confidence in future patterns of change for a given projection.

[*]bsander@ucar.edu
[†]mfwehner@lbl.gov
[‡]reto.knutti@env.ethz.ch





## 2 Introduction

The CMIP5 archive [1] is the most comprehensive collection of climate simulations which has been produced to date. The archive contains simulations from over 25 institutions, some of which submit multiple models - bringing the total number of models in the archive to potentially more than 100 (although many of these are minor variants, and not all models conduct all simulations).

Using this dataset to produce assessments of future climate change involves a number of conceptual challenges. Previous assessments of both the IPCC [2] and the National Climate Assessment in the United States [3] have considered the archive to represent model democracy [4], in that simulations of the future from each model are considered to be equally likely, without accounting for any variation in model skill or for the fact that some models are very similar to other models in the archive, bringing into question the assumption that their simulations can be considered to be independent samples of future behavior.

However, these underlying assumptions have been challenged by a number of studies over recent years. Various studies [5, 6, 7, 8], have pointed out that the ensemble contains demonstrable inter-dependence - where similarities in the spatial biases in model simulations correspond well to expected relationships which one might expect from models from the same institution, or those sharing significant amounts of code. As such, the number of effective models in the archive is likely to be significantly smaller than the number of simulations [9, 10, 7]. The weights should also be representative of the question at hand: skill is not a property of the model per se, but indicative of the ability of a model to project a certain change [11].

In addition, the models that are present in the archive are not equally skillful in representing the present day or past climate [12]. However, it is notably difficult to produce an overall ranking of model performance, given that the conclusion is conditional on both the region and metrics considered [13].

Some studies have suggested methodologies which might be able to address some of these complexities: Bishop et al (2013) [14] proposed a method which produced a set of statistically independent meta models from the original archive, while Sanderson (2015) [7] proposed a method for subsampling the original archive, keeping models which were maximally independent and skillful in reproducing past climate.

In the following study, we present a weighting scheme for use in the 4th National Climate Assessment for the United States. The requirements for this application are somewhat unique - in that a method from the literature cannot be simply taken 'out of the box' from an existing study. Clearly there is a geographical focus: the report itself is focussed on future climate change in the United States, so there is some logic in considering climatological skill which is most relevant to this region. In addition, traceability and simplicity are paramount for this application - so the use of statistical meta-models or narrow subsets of the original archive would not be desirable.

Our methodology is based on the concepts outlined by Sanderson (2015) [7], but instead of deriving a subset, the objective is to produce a single set of model





weights which can be used to combine projections into a weighted mean result, with significance estimates which also treat the weighting appropriately.

The method, ideally, would seek to have two fundamental characteristics. First, if a duplicate of one ensemble member is added to the archive, the resulting mean and significance estimate for future change computed from the ensemble should change as little as possible. Secondly, if a demonstrably poor (for the metrics considered) model is added to the archive, the resulting mean and significance estimates should also change as little as possible.

# 3 Method

## 3.1 Data pre-processing

Our analysis differs in a number of ways from that originally proposed by Sanderson (2015) [7]

- The analysis region contains on the counterterminous United States (CONUS) and most of Canada, constrained by available high resolution observations of daily surface air temperature and precipitation.

- Inter-model distances are computed as simple root mean square differences here, in contrast to the multi-variate PCA used by Sanderson (2015) [7].

- The weights for skill and independence are the final product in this analysis, whereas they only inform the subset choice in the study by Sanderson (2015) [7].

We utilize data for a number of mean state fields, and a number of fields which represent extreme behaviour - these are listed in Table 1. All fields are masked to only include information from the combined CONUS/Canada region. We also consider a selection of models from the CMIP5 archive, listed in Table 2.

## 3.2 Inter-model distance matrix

For each variable, a distance matrix $\delta_v$ is computed between each pair of $N$ total models and between each model and the observed field (such that the observations are treated as an $N + 1^{th}$ model) . Distances are evaluated as the area-weighted root mean square difference over the domain. The matrix is then normalized by the mean inter-model distance, such that for each field in Table 1, there is a $(n_{model} + 1)$ by $(n_{model} + 1)$ matrix representing the pairwise distance between each model (and the observations).

These normalized matrices are then linearly combined, with each line in Table 1 taking equal weight,

$$\delta \quad = \quad \sum_v \delta_v, \qquad (1)$$

to produce the multi-variate distance matrix $\delta$ illustrated in Figure 1.



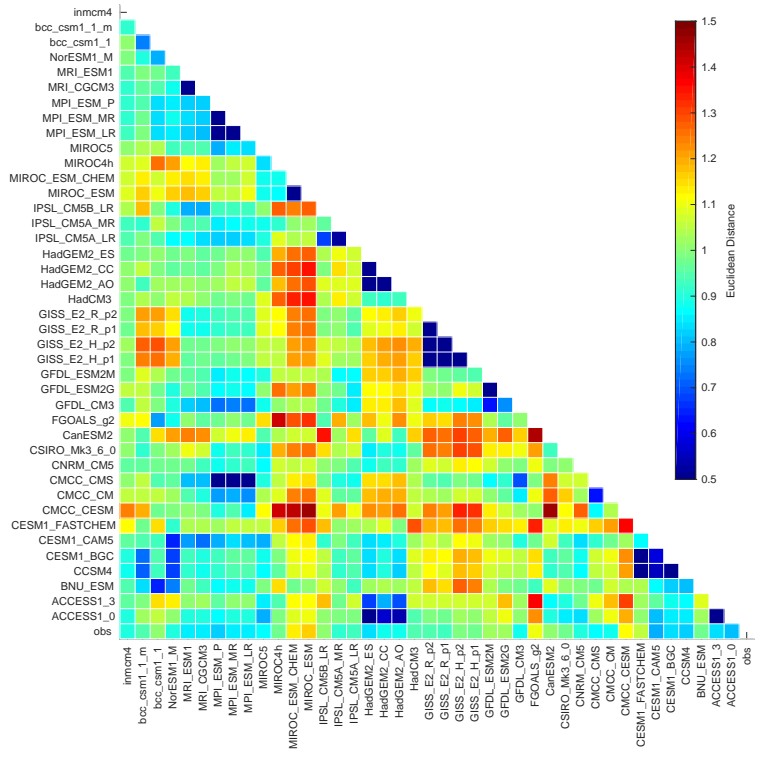

Figure 1: A graphical representation of the inter-model distance matrix for CMIP5 and a set of observed values. Each row and column represents a single climate model (or observation). All scores are aggregated over seasons (individual seasons are not shown). Each box represents a pair-wise distance, where warm colors indicate a greater distance. Distances are measured as a fraction of the mean inter-model distance in the CMIP5 ensemble.



Table 1: Observational Datasets used as observations.

| Field | Description | Source | Reference |
|---|---|---|---|
| TS | Surface Temperature (seasonal) | Livneh, Hutchinson | [15, 15] |
| PR | Mean Precipitation (seasonal) | Livneh, Hutchinson | [15, 15] |
| RSUT | TOA Shortwave Flux (seasonal) | CERES-EBAF | [16] |
| RLUT | TOA Longwave Flux (seasonal) | CERES-EBAF | [16] |
| T | Vertical Temperature Profile (seasonal) | AIRS* | [17] |
| RH | Vertical Humidity Profile (seasonal) | AIRS | [17] |
| PSL | Surface Pressure (seasonal) | ERA-40 | [18] |
| Tnn | Coldest Night | Livneh, Hutchinson | [15, 15] |
| Txn | Coldest Day | Livneh, Hutchinson | [15, 15] |
| Tnx | Warmest Night | Livneh, Hutchinson | [15, 15] |
| Txx | Warmest day | Livneh, Hutchinson | [15, 15] |
| rx5day | seasonal max. 5-day total precip. | Livneh, Hutchinson | [15, 15] |

## 3.3 Model Skill

The RMSE between observations and each model can be used to produce an overall ranking for model simulations of the CONUS/Canada climate (which is illustrated by the overall model-observation distance in Figure 1). Figure 2 shows how this metric is influenced by different component variables.

## 3.4 Independence weights

The inter-model distance matrix is also computed from the inter-model distance matrix $\delta$. For a pair of models $i$ and $j$, we first compute a similarity score $S(\delta_{ij})$ from their pairwise distance $\delta_{ij}$:

$$S(\delta_{ij}) \quad = \quad e^{-\left(\frac{\delta_{ij}}{D_u}\right)^2}, \tag{2}$$

where $D_u$ is the radius of similarity [7], which is a free parameter which determines the distance scale over which models should be considered similar (and thus down-weighted for co-dependence). We show below how an appropriate value can be chosen given prior knowledge about models with known dependencies in the archive.

In limits, two identical models will produce a value of $S(\delta_{ij})$ of 1, and $S(\delta_{ij}) \to 0$ as $\delta_{ij} \to \infty$. A given model $i$'s effective repetition $R_u(i)$ can be calculated by summing the models close by:

$$R_u(i) \quad = \quad 1 + \sum_{j \neq i}^{m} S(\delta_{ij}). \tag{3}$$

Finally, we calculate the independence weight for model $i$ as the inverse of its repetition:



Table 2: Submodel components for the 38 CMIP5 models considered in this study.

| Model | Atmosphere | Land | Ocean | Ice | Source |
|---|---|---|---|---|---|
| NorESM1-ME | CAM4 | CLM4 | MICOM-HAMOCC | CICE | https://verc.enes.org/ISENES2/models/earthsystem-models/ncc/noresm |
| NorESM1-M | CAM4 | CLM4 | MICOM-HAMOCC | CICE | https://verc.enes.org/ISENES2/models/earthsystem-models/ncc/noresm |
| MRI-CGCM3 | MRI-AGCM3 | HAL | MRI.COM3 | | http://www.mri-jma.go.jp/Publish/Technical/DATA/VOL_64/index_en.html |
| MPI-ESM-MR | ECHAM6 | JSBACH | MPIOM | | http://www.mpimet.mpg.de/en/science/models/mpi-esm.html |
| MPI-ESM-LR | ECHAM6 | JSBACH | MPIOM | | https://www.enes.org/models/system-models/mpi-m/mpi-esm |
| MIROC5 | FRCGC-AGCM | MATSIRO | CCSR-COCO | Bitz/Lipscomb | http://journals.ametsoc.org/doi/full/10.1175/2010JCLI3679.1 |
| MIROC4h | FRCGC-AGCM | MATSIRO | CCSR-COCO | Bitz/Lipscomb | http://journals.ametsoc.org/doi/full/10.1175/2010JCLI3679.1 |
| MIROC-ESM-CHEM | FRCGC-AGCM | MATSIRO | CCSR-COCO | Bitz/Lipscomb | http://www.wcrp-climate.org/wgcm/WGCM15/presentations/21Oct/KIMOTO_Japan.pdf |
| MIROC-ESM | FRCGC-AGCM | MATSIRO | CCSR-COCO | Bitz/Lipscomb | http://www.wcrp-climate.org/wgcm/WGCM15/presentations/21Oct/KIMOTO_Japan.pdf |
| IPSL-CM5B-LR | LMDZ (CM4) | ORCHIDEE | NEMO-OPA | NEMO-LIM | http://icmc.ipsl.fr/index.php/icmc-models/icmc-ipsl-cm5 |
| IPSL-CM5A-MR | LMDZ | ORCHIDEE | NEMO-OPA | NEMO-LIM | http://icmc.ipsl.fr/index.php/icmc-models/icmc-ipsl-cm5 |
| IPSL-CM5A-LR | LMDZ | ORCHIDEE | NEMO-OPA | NEMO-LIM | http://icmc.ipsl.fr/index.php/icmc-models/icmc-ipsl-cm5 |
| BCC-CSM1-1-M | BCC-AGCM 2.1 | CLM3 | MOM4 | SIS | http://link.springer.com/article/10.1007%2Fs13351-014-3041-7 |
| BCC-CSM1-1 | BCC-AGCM 2.1 | CLM3 | MOM4 | GFDL_SIS | http://link.springer.com/article/10.1007%2Fs13351-014-3041-7 |
| HadGEM2-ES | HadGAM2 (N96L38) | TRIFFID | HadGOM2 | | http://cms.ncas.ac.uk/wiki/UM/Configurations/HadGEM2 |
| HadGEM2-CC | HadGAM2 (N96L60) | TRIFFID | HadGOM2 | | http://cms.ncas.ac.uk/wiki/UM/Configurations/HadGEM2 |
| HadGEM2-AO | HadGAM2 (N96L38) | MOSES2 | HadGOM2 | | http://cms.ncas.ac.uk/wiki/UM/Configurations/HadGEM2 |
| GISS-E2-R | GISS | GISS | Russell | Russell | http://data.giss.nasa.gov/modelE/ar5/ |
| GISS-E2-H | GISS | GISS | HYCOM | HYCOM | http://data.giss.nasa.gov/modelE/ar5/ |
| GFDL-ESM2M | GFDL-AM2.1 | LM3 | MOM4.1 | SIS | http://cms.ncas.ac.uk/wiki/UM/Configurations/HadGEM2 |
| GFDL-ESM2G | GFDL-AM2.1 | LM3 | GOLD | SIS | http://www.gfdl.noaa.gov/earth-system-model |
| GFDL-CM3 | GFDL-AM3 | LM3 | MOM4.1 | SIS | http://www.gfdl.noaa.gov/earth-system-model |
| FGOALS-g2 | GAMIL 2.0 | CLM3 | LICOM2 | CICE4.LASG | http://link.springer.com/article/10.1007%2Fs00376-012-2140-6 |
| CanESM2 | AGCM4 | CLASS | NCAR | SIS | http://journals.ametsoc.org/doi/pdf/10.1175/JCLI-D-11-00715.1 |
| CSIRO-Mk3-6-0 | Gordon | CABLE | MOM2.2 | SIS | http://www.bom.gov.au/amoj/docs/2013/jeffrey_hres.pdf |
| CNRM-CM5 | ARPEGE-Climate | ISBA | NEMO-OPA | GELATO | http://www.cnrm-game.fr/spip.php?article126&lang=en |
| CMCC-CMS | ECHAM5 | SILVA | OPA8.2 | LIM | http://www.wcrp-climate.org/wgcm/WGCM16/Bellucci_CMCC.pdf |
| CMCC-CM | ECHAM5 | SILVA | OPA8.2 | LIM | http://www.cmcc.it/models/cmcc-cm |
| CMCC-CESM | ECHAM5 | SILVA | OPA8.2 | LIM | http://www.cmcc.it/models/cmcc-cm |
| CESM1-CAM5 | CAM5 | CLM4 | POP2 | CICE4 | https://www2.cesm.ucar.edu/models |
| CESM1-FASTCHEM | CAM5 | CLM4 | POP2 | CICE4 | https://www2.cesm.ucar.edu/models |
| CESM1-BGC | CAM4 | CLM4 | POP2 | CICE4 | https://www2.cesm.ucar.edu/models |
| CCSM4 | CAM4 | CLM4 | POP2 | CICE4 | https://www2.cesm.ucar.edu/models |
| BNU-ESM | CAM3.5 | CLM/BNU | MOM4.1 | CICE4.1 | http://www.wcrp-climate.org/wgcm/WGCM15/presentations/21Oct/WANG_WGCM.pdf |
| BCC-CSM1-1-M | BCC-AGCM 2.1 | CLM3 | MOM4 | SIS | http://link.springer.com/article/10.1007%2Fs13351-014-3041-7 |
| BCC-CSM1-1 | BCC-AGCM 2.1 | CLM3 | MOM4 | GFDL_SIS | http://link.springer.com/article/10.1007%2Fs13351-014-3041-7 |
| ACCESS1-3 | UKMO_GA1.0 | CABLE_v1.8 | MOM4.1 | CICE4.1 | https://wiki.csiro.au/display/ACCESS/Home |
| ACCESS1-0 | HadGEM2_r1.1 | MOSES | MOM4.1 | CICE4.1 | http://www.cawcr.gov.au/publications/technicalreports/CTR_059.pdf |



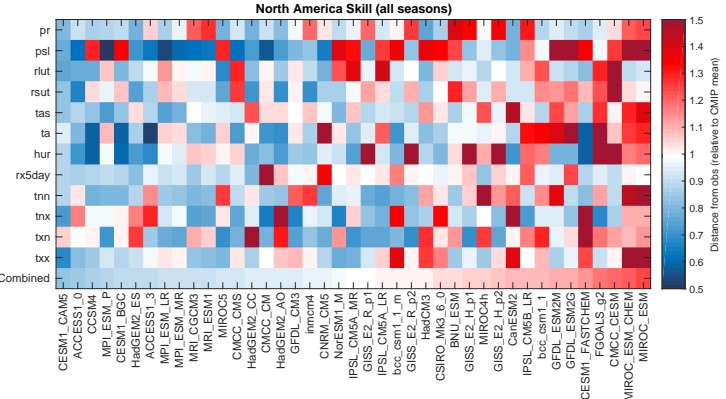

Figure 2: A graphical representation of the model-observation distance matrix for a number of variables, illustrating how different biases combine to produce the overall model-observation distance in Figure 1. Each column represents a single climate model, and rows represent the different observation types in Table 1. Distances along each row are normalized, such that the mean model has a distance of 1 to the observations. CMIP5 Models are sorted by their combined skill as shown in the bottom row.

$$w_u(i) \quad = \quad (R_u(i))^{-1}. \tag{4}$$

Figure 3 shows the dependence of the independence weights on $D_u$ for a number of different models. $D_u$ is sampled by considering the distribution of inter-model distances $\delta$, and sampling by percentiles $\sigma_u$ the smallest inter-model distances in the archive.

As points of reference, we consider some models from the archive known to have no obvious duplicates (HadCM3 and INMCM), which should not be significantly down-weighted by the method. We also consider some models where there numerous known closely related variants submitted from MIROC, MPI and GISS. It is desirable to choose a value of $D_u$ which produces a weight of approximately $1/n$ where $n$ is the number of variants submitted.

Hence, by inspection of Figure 3, we take $D_u$ as 0.48 times the distance between the best performing model and observations in the CMIP5 archive, which produces approximately the desired weighting characteristics in these cases where we have a reasonable expectation of what the true model replication is in the archive.

The methodology described above assumes each model has submitted only one simulation to the archive, but the method is robust to the inclusion of multiple initial condition members from each model. If $D_u$ is chosen such that





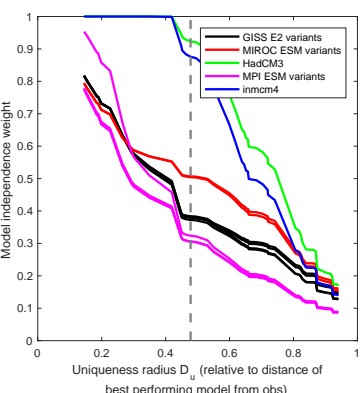

Figure 3: Model independence weights ($w_u$) as a function of the radius of interdependence $D_u$, plotted for a number of models and groups of models in the CMIP5 archive. The vertical line shows the value used in NCA4.

structurally similar ensemble members, then $w_u$ will appropriately allocate a
fractional weight to each initial condition ensemble member. In the case of
NCA4, extreme value statistics were only available for a single instance of each
model, hence initial condition ensembles were not considered.

## 3.5  Skill weights

The RMSE distances between each model and the observations are used to
calculate skill weights for the ensemble. The skill weights represent the clima-
tological skill of each model in simulating the CONUS/Canada climate, both in
terms of mean climatology and extreme statistics. The skill weighting $w_q(i)$ for
model $i$ is calculated as in [7]:

$$w_q(i) \;=\; e^{-\left(\frac{\delta_{i(obs)}^{20c}}{D_q}\right)^2}, \tag{5}$$

where $\delta_{i(obs)}^{20c}$ is the sum of the normalized RMSE differences over all variables,
between each model and the observations, and $D_q$ is the radius of model quality
[7] which determines the degree to which models with a poor climatological
simulation should be downweighted. As such, a very small value of $D_q$ will
allocate a large fraction of weight to the single best performing model in the
archive (as assessed by the climatological skill). Equally, as $D_q \to \infty$, the
multi-model average will tend to the non skill-weighted solution.
An overall weight is then computed as the product of the skill weight and
the independence weight.





$$w(i) \quad = \quad w_u(i)w_q(i) \tag{6}$$

We determine an appropriate value for $D_q$ by considering both the skill of the
weighted average in reproducing observations, and also by conducting perfect
model simulations with the CMIP5 ensemble. In Figure 4(a), we show that the
use of relatively strong weighting (where the $D_q$ is 50 percent of the distance
between the best performing model and the observations) produces the weighted
climatological average with the lowest error.

However, a more skillful representation of the present-day state does not
necessarily translate to a more skillful projection in the future. In order to assess
whether our metrics improve the skill of future projections at all, we consider
a perfect model test where a single model is withheld from the ensemble and
then treated as truth.

However, such a test can be over-confident because when some models are
treated as truth, there remain close relatives of that model in the archive which
would be given a high skill weight and would inflate the apparent skill of the
metric in predicting future climate evolution. To partly address this, we conduct
our perfect model study with a subset of the CMIP5 archive which excludes
obvious near relatives of the chosen 'truth' model. We achieve this by excluding
any model which lies closer to the 'truth' model than the distance between the
best performing model and the observations in the inter-model distance matrix
$\delta$. The excluded model pairs for the perfect model test are illustrated in Figure
5.

Once the obvious duplicates have been removed, we can test the ability of
the chosen multivariate climatological metrics to increase skill in the simulation
of the out of sample model's future. We do this in two ways: in the first
case, we consider the RMSE of the weighted multi-model mean projection of
each out of sample model's projection of annual mean gridded temperature and
precipitation change at the end of the 21st century under RCP8.5. This is
expressed as a fraction of the RMSE one would obtain with a simple mean of
the remaining models (again, excluding the obvious duplicates). This process is
repeated for each model in the archive, after which the results are averages and
plotted in Figure 4(b), where the optimum value of $D_q$ for the reproduction of
future temperature and precipitation change is approximately 70 percent of the
distance between the best performing model and observations, for which there is
a 9-10 percent reduction in RMSE compared the unweighted case. This suggests
that in the perfect model study, some skill weighting based on climatological
performance can improve the mean projection of future change.

Finally, we test whether skill-weighting the ensemble increases the chances
of the truth lying outside of the distribution of projections suggested by the
archive. For Figure 4(c), we consider the ensemble projected values for future
temperature and precipitation at each gridcell, using the combined skill and
independence weight (with the perfect model treated as observations) to define
a likelihood distribution for future change. We show the average fraction of



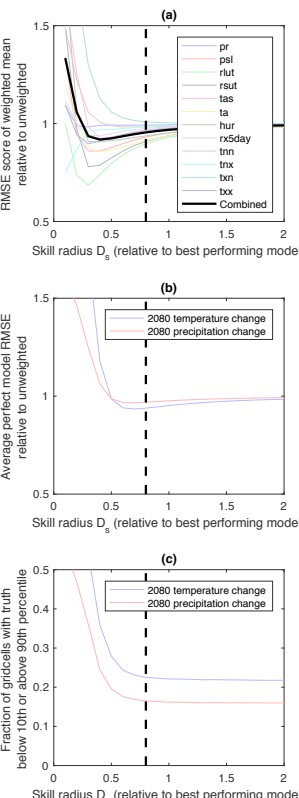

Figure 4: Subplots are functions of $D_q$, the radius of model quality (all figures take a value of $D_u$ corresponding to the 1.5th percentile of the inter-model distance distribution). Subplot (a) shows the RMSE of the weighted multi-model mean compared with observations, relative to the non skill-weighted multi-model mean. Subplot (b) shows the average RMSE of future annual mean gridded temperature change projections in 2080-2100 (relative to 1980-2000) under RCP8.5 for an out-of sample model taken to represent truth (with obvious replicates removed from the ensemble). Subplot (c) shows the average fraction of grid-cells for which the out-of sample 'perfect model' projections lie below the 10th or above the 90th percentile of the inferred weighted distribution.





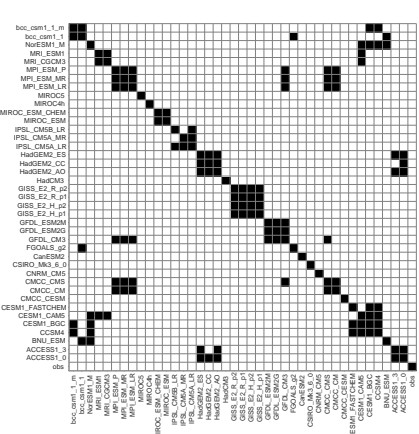

Figure 5: A graphical representation of models which are excluded from the remaining ensemble in the perfect model test when each model in turn is treated as truth. Cells in black represent models which are closer to each other than the best performing model in the archive is to observations.





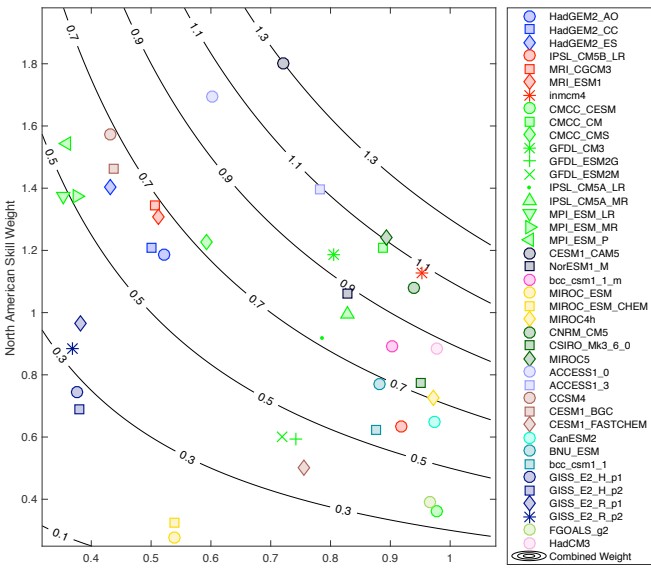

Figure 6: Model skill and independence weights for the CMIP-5 archive evaluated over the CONUS/Canada domain. Contours show the overall weighting, which is the product of the two individual weights.

grid-cells where the chosen perfect model projected value for temperature or precipitation change lies above the 90th or below the 10th percentile of that distribution. If the likelihood distribution is representative, one would expect 20 percent chance that the perfect model lies in this range. However, if this value increases, it indicates that the weighting is too strong and the weighting is producing an under-dispersive distribution.

Figure 4(c) shows that for values of $D_q$ of less than 80 percent of the distance between the best performing model and observations, there is some increased risk of the ensemble being under-dispersive. As such, this is a justifiable value to retain - there is still a demonstrable increase in the out-of-sample skill of the future projection in the perfect model tests, with a minimal risk of an under-dispersive distribution.

Using the values of $D_q$ and $D_u$ defended in this section, we illustrate skill, independence and combined weights for the CMIP5 archive in Figure 6 and in Table 3.



| | Uniqueness weight | Skill Weight | Combined |
|---|---|---|---|
| ACCESS1-0 | 0.60 | 1.69 | 1.02 |
| ACCESS1-3 | 0.78 | 1.40 | 1.09 |
| BNU-ESM | 0.88 | 0.77 | 0.68 |
| CCSM4 | 0.43 | 1.57 | 0.68 |
| CESM1-BGC | 0.44 | 1.46 | 0.64 |
| CESM1-CAM5 | 0.72 | 1.80 | 1.30 |
| CESM1-FASTCHEM | 0.76 | 0.50 | 0.38 |
| CMCC-CESM | 0.98 | 0.36 | 0.35 |
| CMCC-CM | 0.89 | 1.21 | 1.07 |
| CMCC-CMS | 0.59 | 1.23 | 0.73 |
| CNRM-CM5 | 0.94 | 1.08 | 1.01 |
| CSIRO-Mk3-6-0 | 0.95 | 0.77 | 0.74 |
| CanESM2 | 0.97 | 0.65 | 0.63 |
| FGOALS-g2 | 0.97 | 0.39 | 0.38 |
| GFDL-CM3 | 0.81 | 1.18 | 0.95 |
| GFDL-ESM2G | 0.74 | 0.59 | 0.44 |
| GFDL-ESM2M | 0.72 | 0.60 | 0.43 |
| GISS-E2-H-p1 | 0.38 | 0.74 | 0.28 |
| GISS-E2-H-p2 | 0.38 | 0.69 | 0.26 |
| GISS-E2-R-p1 | 0.38 | 0.97 | 0.37 |
| GISS-E2-R-p2 | 0.37 | 0.89 | 0.33 |
| HadCM3 | 0.98 | 0.89 | 0.87 |
| HadGEM2-AO | 0.52 | 1.19 | 0.62 |
| HadGEM2-CC | 0.50 | 1.21 | 0.60 |
| HadGEM2-ES | 0.43 | 1.40 | 0.61 |
| IPSL-CM5A-LR | 0.79 | 0.92 | 0.72 |
| IPSL-CM5A-MR | 0.83 | 0.99 | 0.82 |
| IPSL-CM5B-LR | 0.92 | 0.63 | 0.58 |
| MIROC-ESM | 0.54 | 0.28 | 0.15 |
| MIROC-ESM-CHEM | 0.54 | 0.32 | 0.17 |
| MIROC4h | 0.97 | 0.73 | 0.71 |
| MIROC5 | 0.89 | 1.24 | 1.11 |
| MPI-ESM-LR | 0.35 | 1.38 | 0.49 |
| MPI-ESM-MR | 0.38 | 1.37 | 0.52 |
| MPI-ESM-P | 0.36 | 1.54 | 0.56 |
| MRI-CGCM3 | 0.51 | 1.35 | 0.68 |
| MRI-ESM1 | 0.51 | 1.31 | 0.67 |
| NorESM1-M | 0.83 | 1.06 | 0.88 |
| bcc-csm1-1 | 0.88 | 0.62 | 0.55 |
| bcc-csm1-1-m | 0.90 | 0.89 | 0.80 |
| inmcm4 | 0.95 | 1.13 | 1.08 |

Table 3: Uniqueness, Skill and Combined weights for CMIP5 for the CONUS/Canada domain





## 4   Gridded application

Once derived, the skill and independence weights can be used to to produce weighted mean estimates of future change, as well as confidence estimates for those projections. To illustrate this, we modify the significance methodology from the 5th Assessment Report of the IPCC [2], such that:

- Stippling - large changes where the weighted multimodel average change is greater than double the standard deviation of the 20 year mean from control simulations runs and 90 percent of the weight corresponds to changes of the same sign.

- Hatching - No significant change where the weighted multimodel average change is less than the standard deviation of the 20 year means from control simulations runs.

- Blanked out - Inconclusive where the weighted multimodel average change is greater than double the standard deviation of the 20 year mean from control runs and less than 90 percent of the weight corresponds to changes of the same sign.

Following the protocol of [2], the standard deviation of the 20 year mean from control simulations is derived using the 'picontrol' simulations in CMIP5. We consider all simulations with a length of 500 years or longer, and discard the first 100 years. The remaining time period is broken into consecutive 20 year periods, and the estimate of control variability for each model is taken as the standard deviation of the 20 year periods. This process is repeated for all models with an appropriate simulation. Finally, the standard deviations are averaged over all models to produce the final estimate for the standard deviation of the 20 year mean from the control simulations.

In order to adapt this methodology to a weighted ensemble, we need to apply the weights both to the mean estimate and the significance estimates.

To calculate the weighted average, each model is associated with a weight (e.g. from table 3). The weights must be normalized, and the weighted average $p$ at each gridcell is:

$$p = 1/n \sum_{1}^{n} w(i)p(i) \tag{7}$$

where $n$ is the number of models, $w(i)$ is the weight of model $i$ and $p(i)$ is the projected value from model $i$.

Therefore, the significance test is very similar to the IPCC case: if the weighted average exceeds double the control standard deviation, it is a significant change and if it is less than the standard deviation it is not significant.

Sign agreement is slightly modified from the IPCC case - rather than assessing the number of models exhibiting the same sign of change, we consider





the fraction of the weight exhibiting the same sign of change, $f$. This can be expressed as:

$$f = |1/n \sum_{1}^{n} w(i)\text{sign}(p(i))|, \qquad (8)$$

for any given set of projections $p$.

We illustrate the application of this method to future projections of temperature and precipitation change under RCP8.5 in Figures 7 and 8 which show the mean projected quantities as well as the 10th and 90th percentiles of the weighted distribution of change at the gridcell level. In both cases, the weighting has only a subtle effect on the mean projection, but serves to slightly constrain the range of response at a given gridcell. In Section 5, we discuss how more aggressive or targeted weighting can have a greater potential effect.

# 5 Sensitivity Studies

The parameter choices for $D_q$ and $D_u$ utilized in Section 3, as well as the choice of metrics and the domain were considered appropriate for the specific application of the US National Assessment, where it was desirable to have a single set of weights used for a number of applications. However, in a more general sense, we consider here how different choices may impact the results of weighted analyses, and how the researcher should consider weighting in more targeted (or more global) applications. We briefly consider how the sensitivities of the method to different choices.

## 5.1 Spatial Domain

In the case of NCA4, the strategy was to produce multi-variate metrics which were specific to CONUS/Canada. However, there is an argument that there are aspects of non-local climatology which would ultimately impact the domain of interest (through their influence on global climate sensitivity, for example).

In Figure 9(a-e), we consider the RMSE metrics for both the US and the entire global domain. In this comparison, it is shown that there is a relatively poor correlation between model skill evaluated over CONUS/Canada and globally for any individual metric, however, when individual metrics are combined into a multivariate climate (the approach used in Section 3), there is a correlation of 0.89 between the regional and local metrics. As such, the final weighting for NCA4 would not be highly sensitive to using global rather than CONUS/Canada metrics, but a study using a more restrictive set of variables to assess model quality could potentially be sensitive to domain choice.

## 5.2 Skill weighting strength

The strength of the skill weighting corresponds to the parameter $D_s$ in Section 3. For the purpose of NCA4, a conservative value was chosen to minimize the potential for overconfidence in future projections from the weighted ensemble.





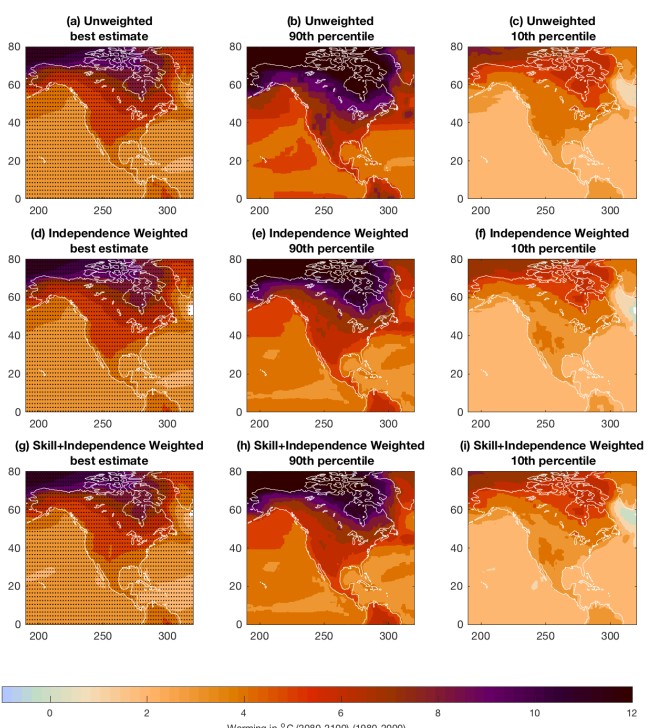

Figure 7: Projections of mean temperature change over CONUS/Canada in 2080-2100, relative to 1980-2000 under RCP8.5. (a-c) show the simple un-weighted CMIP5 multi-model average, 90th percentile of warming and 10th percentile of warming using the significance methodology from [2], (d-f) show the weighted results as outlined in section 4 for models weighted by uniqueness only and (g-i) show weighted results for models weighted by both uniqueness and skill.





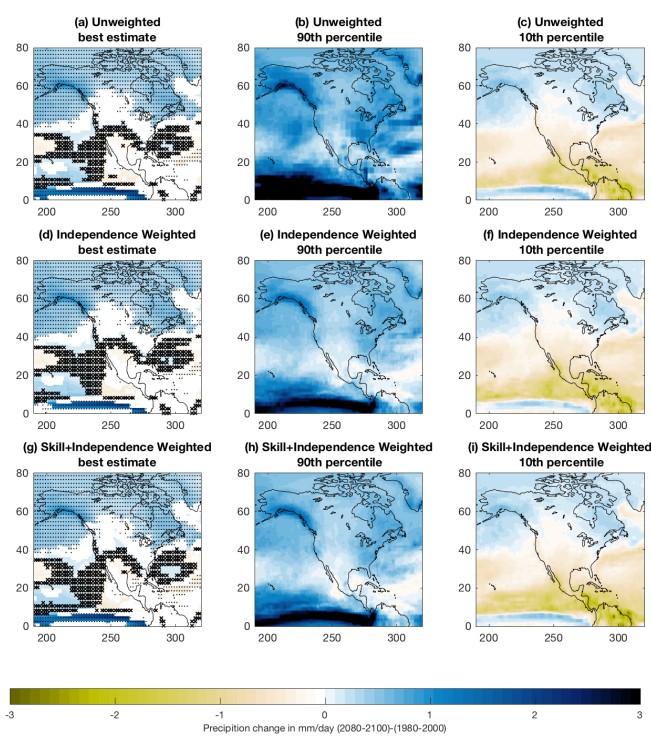

Figure 8: As for Figure 7, but for future mean precipitation change under RCP8.5.





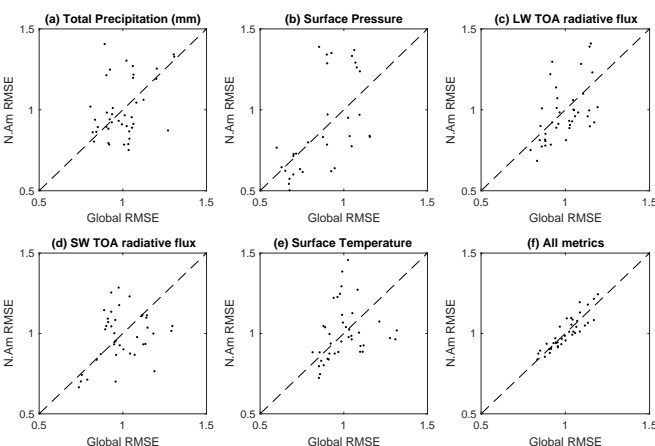

Figure 9: A series of plots showing Root Mean Square Errors evaluated over the CONUS/Canada domain as a function of errors assessed over the global domain. Each point corresponds to a single model in the CMIP5 archive. Plots are shown for some individual fields (a-e) and (f) RMSE averaged over all 12 available fields listed in Figure 2.



This resulted in only very subtle changes in gridded temperature and precipitation projections for the future (although there are some noticeable differences in the uncertainty range, see Figures 7 and 8).

However, here we consider the impact on temperature projections if a more aggressive weighting strategy were used. In Figure 10(a), we show the sensitivity of global mean temperature change under RCP8.5 as a function of the skill radius. The default value of $D_s = 0.8$ produces a small decrease in projected 2080-2100 global mean temperature increase (a warming of 3.7K above 1980-2000 levels, compared to the non-skill weighted case of 3.9K, Figure 10(d)).

As $D_s \rightarrow 0$, the fraction of the percent of the models associated with 90 percent of the weight decreases, and more weight is placed upon the models with higher combined skill scores in Figure 2. If a value of $D_s = 0.4$ is used, 90 percent of the model weight is allocated to just 40 percent of models, and the projected warming is decreased further to 3.45K (Figure 10(c)). However, if $D_s$ is reduced further to 0.1, such that 90 percent of weight is placed on only the top 5 percent of models (which corresponds to only 2 models: CESM1-CAM5 and ACCESS1.0), the weighted warming estimate is higher than the unweighted case at 4.1K (Figure 10(b)).

Hence, we find that although a the skill weighting as used in NCA4 has only a subtle effect on projected temperatures compared to the unweighted case, there is a demonstrable effect when stronger weights are utilized, but there is an increased risk of the weighted ensemble being underdispersive (Figure 4(c)). For very aggressive weighting, projections differ significantly from the unweighted case but the resulting projection is effectively governed by only the best performing few models, such agressive weighting in the perfect model test was found to result in a less skillful projection (Figure 4(b)).

## 5.3 Univariate weighting

The requirements for NCA4 were such that a single set of weights should be used for the entire report. However, for some application it might be desirable to taylor a set of weights to optimally represent a particular process or projection. Here, we consider how using weights assessed on precipitation climatology alone could change the result of the projection. The precipitation weighted case is formulated identically to the multivariate case but distances are computed using RMS differences over the mean precipitation field (over the CONUS/Canada domain) only; the selection of $D_s$ is set to 0.8 times the distance of the best performing model, and $D_u$ is taken the 1.5th percentile of the inter-model distance distribution as in the multivariate case.

Figure 11(a) shows the distribution of changes in grid-level precipitation for the late 21st century under RCP8.5. It is notable that there is negligible difference between the mean precipitation changes in the unweighted case and the multi-variate weighted case, but in the precipitation only case there is an increase in regions exhibiting a large drying trend. This implies that a multi-variate metric has little constraint on precipitation change, but a more targeted metric could potentially identify regions which might exhibit extreme drying



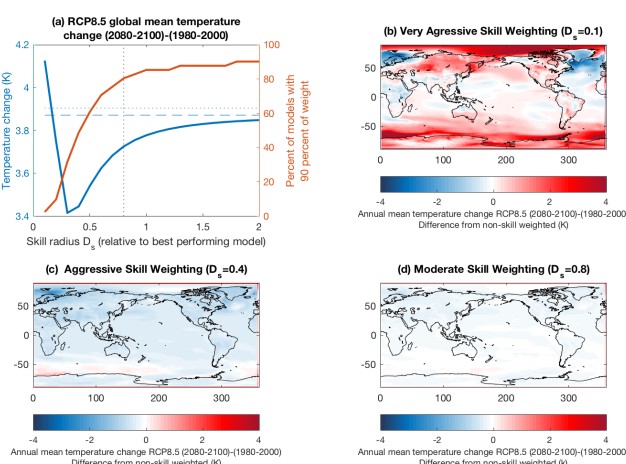

Figure 10: A plot showing the effect of skill weighting strength on global temperature projections. Subplot (a) shows global mean temperature increase for 2080-2100 under RCP8.5 as a function of the skill radius $D_s$ (blue curve), as well as the fraction of models with 90 percent of the allocated weight (red curve). Subplots (b-d) show projected mean temperature maps for 3 cases of $D_s$=0.1 (b), 0.4 (c) and 0.8 (d).





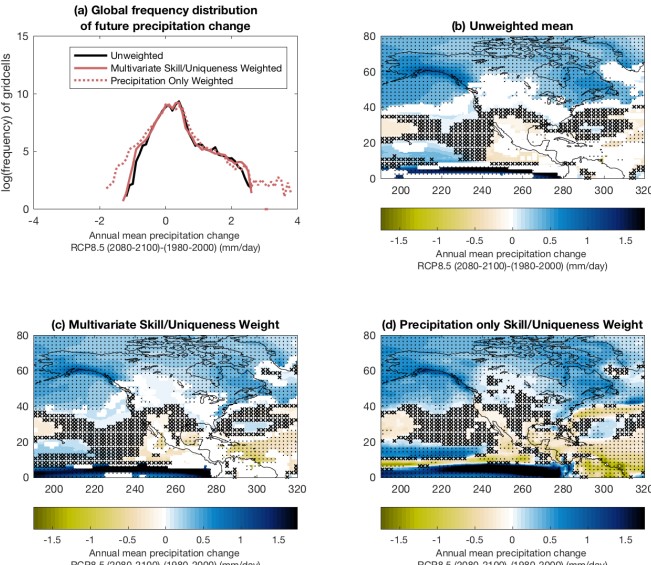

Figure 11: Distribution of changes in grid-level precipitation for the late 21st century under RCP8.5. (a) shows the distribution for the mean (black) or weighted by all variables (red solid) and weighted by precipitation only (red dotted) projection of annual precipitation under RCP8.5. (b-d) show maps of precipitation change in the style of Figure 8 for each weighting case.

in the future (just as each individual model exhibits some regions of extreme
drying, but the lack of agreement amongst models on where those regions are
causes the multi-model mean to lack any such behavior).

We can illustrate this behavior by considering the spatial pattern of precip-
itation change in the three cases, using unweighted(Figure 11(b)), multivariate
weighted (Figure 11(c) as in Figure 8) or weighted using only the climatological
precipitation only (Figure 11(d)). In the unweighted case, large fractions of the
continental US show disagreement in the sign of precipitation change. Much of
the midwest, northwest and southwest Canada for example are colored white
indicating that models disagree on the sign of change, and drying in the south-
west is not significant. A multivariate weighting makes little difference to this
assessment; there is some indication that increased precipitation in the northern
US is more likely - but changes still fail to be significant.

A precipitation-based metric, however, seems to make a noticeable difference
to the confidence associated with the weighted projection. There is now clear





and significant increases in precipitation in the northern part of the US, and significant increases in the northeast. There is also more clearly defined drying along the west coast and significant drying over the northern Amazon which was not evident in the unweighted or multivariate case.

Hence, it seems that there is potential to constrain the spatial patterns of fields which show significant spatial heterogeneity across the multi-model archive by considering targeted metrics which might be more directly informative to relevant processes for that particular projection. One must be cautious because as noted in Section 5.1, because individual metrics are more susceptible to domain choice than the multivariate case, and so such a targeted constraint must be thoroughly investigated before application in a general assessment. However, this is a potential line of investigation which would be worthy of future study.

# 6 Summary and Discussion

This study has discussed a potential framework for weighting models in a structurally diverse ensemble of climate model projections, accounting for both model skill and independence. The parameters of the weighting in this case were optimized for using the CMIP5 ensemble in the fourth National Climate Assessment for the United States (NCA4); an application which required a weighting strategy targeted towards a particular region (CONUS/Canada), with a single set of weights which could be applied to a diverse range of projections.

The solution proposed in this study adapted the logic first discussed in the context of model sub-selection in Sanderson et al (2015) [7], and applied it to a continuous weighting scheme. Weights were formulated on the basis of skill and uniqueness, where skill was assessed by considering the climatological bias averaged over a diverse set of variables, and uniqueness was assessed by constructing an inter-model distance matrix from the same set of variables and down-weighting models which lie in each others' immediate vicinity.

A single set of weights constructed for NCA4, using a multi-variate climatological skill metric and a limited domain size. Two parameters must be determined for the weighting algorithm; a radius of model skill and one of similarity. The former was calibrated by considering a perfect model test where a single model is treated as truth and its historical simulation output is treated as observations, immediate neighbors of the test model are removed from the archive and the remaining models are used to conduct tests which assess skill in reconstructing past and future model performance, as well as assessing the risk of producing an underdispersive ensemble which fails to encompass the perfect future projection at a given grid point. Using these three tests, we take a conservative choice for model weighting which minimizes the risk of under-dispersion (i.e. the risk that the real world might lie outside the entire weighted distribution of projections at a given gridpoint).

The similarity parameter is calculated in a qualitative fashion by considering known cases where models are known to be unique, or where there is a known set of closely related models. The parameter is adjusted such that the known-unique





<sub>395</sub> models are given a weight of near unity, and the models with $n$ near-identical
<sub>396</sub> versions are each given a weight of approximately $1/n$.

<sub>397</sub> The requirements of a large assessment place constraints on the choice of
<sub>398</sub> parameters for this analysis. Logistical considerations imply that only one set
<sub>399</sub> of weights can be constructed, and the broad readership and high stakes of the
<sub>400</sub> assessment mean that any risk of under-dispersion of projected future climate is
<sub>401</sub> unacceptable for this application. These constraints dictate that only a moder-
<sub>402</sub> ate weighting of model skill is used, where 90 percent of the weight is allocated
<sub>403</sub> to 80 percent of models. This, unsurprisingly, creates only a modest change in
<sub>404</sub> mean projected results and only a small reduction in uncertainty. A stronger
<sub>405</sub> skill weighting is shown to have a more significant effect on projected changes,
<sub>406</sub> but with the risk of increased under-dispersion.

<sub>407</sub> In addition, there exists a weak trade-off between model skill and model
<sub>408</sub> uniqueness in the CMIP5 ensemble; models which are demonstrably high per-
<sub>409</sub> forming also tend to be the ones with the most near replicates in the archive. As
<sub>410</sub> such, there is a compensating effect of the skill and uniqueness components of
<sub>411</sub> the weighting algorithm, which tends to mute the effect of the overall weighting
<sub>412</sub> when compared to the unweighted case. In other words, the unweighted CMIP5
<sub>413</sub> ensemble is in fact already a skill weighted ensemble to some degree.

<sub>414</sub> However, although this tradeoff is evident in the CMIP5 archive, there is
<sub>415</sub> no guarantee that such a tradeoff is a justification for using an unweighted
<sub>416</sub> average in future versions of the CMIP archive. A single, highly replicated
<sub>417</sub> but climatologically poor model present in a future version of the archive could
<sub>418</sub> significantly bias the simple multi-model mean of a climatological projection. As
<sub>419</sub> such, it is desirable to have a known and tested weighting algorithm in place to
<sub>420</sub> produce robust projections in the case of highly replicated, or very poor models.

<sub>421</sub> Beyond the single set of weights produced for NCA4, the basic structure
<sub>422</sub> outlined in this study can be used to produce a more targeted weighting for
<sub>423</sub> a particular projection. Our provisional results suggest that targeted weights
<sub>424</sub> could potentially yield more confidence in projections if only a limited set of
<sub>425</sub> relevant projections are included, especially in fields where projections exhibit
<sub>426</sub> high degrees of structural diversity within the archive. This tailored weighting
<sub>427</sub> approach, however, presents risks which necessitate further study - our sensi-
<sub>428</sub> tivity studies suggest that multi-variate metrics are more robust to changes in
<sub>429</sub> spatial domain than targeted metrics, and the exact choice of metrics which
<sub>430</sub> should be used to best constrain a particular projection is not trivial matter.

<sub>431</sub> With this in mind, we propose that future studies should further investi-
<sub>432</sub> gate how selection of physically relevant variables and domains should be used
<sub>433</sub> to optimally weight projections of future climate change- and that individual
<sub>434</sub> projections will need careful consideration of relevant processes in order to for-
<sub>435</sub> mulate such metrics. Confidence in such weighting approaches is highest if there
<sub>436</sub> are well understood underlying processes that explain why the chosen metric
<sub>437</sub> constrains the projection. Until then, we have presented a provisional and con-
<sub>438</sub> servative framework which allows for a comprehensive assessment of model skill
<sub>439</sub> and uniqueness from the output of a multimodel archive when constructing
<sub>440</sub> combined projections from that archive. In so doing, we come to the reassuring



conclusion that for this particular application (i.e., domain and variables) the results which would be inferred from treating each member of the CMIP5 as an independent realization of a possible future are not significantly altered by our weighting approach. However, by establishing a framework, we make the first tentative steps away from simple model democracy in a climate projection assessment, leaving behind a strategy which is not robust to highly unphysical or highly replicated models of our future climate.

# 7   Code availability

Complete MATLAB code for the analysis conducted in this manuscript is provided. All CMIP5 data used in this analysis is downloadable from the Earth System Grid (https://pcmdi.llnl.gov/projects/esgf-llnl/).

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
