# Peer review of "Skill and independence weighting for multi-model assessments"

_Geoscientific Model Development, 2016_

## Referee Comment (RC1) · Anonymous Referee #1 · 27 Feb 2017

The authors present a weighting scheme which, in their words, "considers both skill in the climatological performance of models over North America as well as the interdependency of models arising from common parameterizations or tuning practises". The two components of the weighting scheme are presented and developed separately, and are found to compensate to some extent for each other: models that are weighted higher for performance tend to be down weighted for replication.

The main limitation of the manuscript in my view is the primarily heuristic nature of the weighting schemes, which are at best partially justified. The introduction l73-78 sets out "two fundamental characteristics" of the scheme which are probably uncontroversial but which are not sufficient to narrow down the nature of the weighting scheme very much. I would however suggest that "relatively poor" would be more precise than the stated "demonstrably poor".

Taking performance weighting first, there is a substantial literature on this, albeit perhaps with limited results. Methods based on Bayesian Model Averaging (e.g. Hoeting et al 1999) have perhaps the strongest theoretical justification, but other approaches have also been presented (such as the "reliability ensemble averaging" approach of Giorgi and Means 2002). Olson et al 2016(a,b) present some recent applications of BMA to regional projections which seem highly relevant. I would ask the authors to consider whether their performance weights can be considered as Bayesian likelihoods, that is to say, is there an underlying statistical model which would result in this weighting scheme? If not, would it be worth changing to a more transparently presented and explained model, perhaps one which has been more widely applied and tested? Of course any statistical method will necessarily rest on a number of assumptions and simplifications which may not be easily justified, but at least these could be presented explicitly. For example, while the distance factor Dq is considered as a tunable factor here, there is also the use of an exponential function which defines the weights, for which no explanation is given. Even without changing the overall structure of the weighting function, increasing the exponent from its value of 2 would result in a sharper cliff-edge at which weights drop from 1 to 0, and alternatively a lower exponent would result in a much more gradual change with weights more similar across the models. Is there a particular reason for the choices made here?

Now moving on to the question of model independence, which here seems to be used to mean model output difference (as measured by a metric on output fields). The functional choice for the weighting again seems rather arbitrary. Since the goal of the parameter tuning seems to be to match the authors' beliefs that various models are replicated a particular numbers of times, is there a reason to use a function - which can only provide an approximation to this prior belief - rather than just use the authors' own judgements instead? For example a weight of 1/4 say could be applied to the GISS models directly, rather than trying to obtain a value close to this by tuning a single parameter. The choice of a fitted function seems to provide only a very thin veneer of objectivity to this subjective choice.

Despite these comments, I have no particular beef with the framework that has been presented - it does not look wrong or silly in any obvious way - but I also don't feel like I have been given any particular reason for using it. As outlined above, several of the numerous choices made don't appear to be that well justified. The tuning parameters do appear to have been selected sensibly, but this is only the last step after the creation of a structure that doesn't seem well supported.

A number of typos:

273-4 We briefly consider how the sensitivities of the method to different choices.

322 taylor/tailor

Fig 4 caption "1.5th percentile" really?

Giorgi, F., & Mearns, L. (2002). Calculation of average, uncertainty range, and reliability of regional climate changes from AOGCM simulations via the "reliability ensemble averaging"(REA) method, J. Climate 15(10), 1141–1158. Hoeting, J., Madigan, D., Raftery, A., & Volinsky, C. (1999). Bayesian model averaging: A tutorial. Statistical Science, 14(4), 382–401. Olson, R., Fan, Y. and J. P. Evans (2016), A simple method for Bayesian model averaging of regional climate model projections: Application to southeast Australian temperatures, Geophysical Research Letters, vol. 43, no. 14, pp. 7661-7669 Olson, R., J. P. Evans, A. Di Luca and D. ArguÌĹeso (2016) The NARCliM project: model agreement and significance of climate projections. Climate Research, 69, 209-227.

---

## Referee Comment (RC2) · C. H. Bishop (Referee) · 6 Mar 2017

General Comments

Climate models mathematically express our understanding of the factors governing weather and climate. The CMIP5 multi-model archive pools together climate projections from leading weather and climate forecasting centers. Taken together, they represent an unprecedented concentration of mankind's knowledge and climate prediction capabilities. However, not all of the models are of the same quality and some of the models are very similar to each other. How then could the information from the individual models be combined to minimize the uncertainty in climate change uncertainty? One option is simply to take the unweighted sample mean of the ensemble of predictions. An alternative option pursued by Sanderson and co-authors in this paper is

that of assigning weights to the ensemble members based on a combination of their skill and uniqueness characteristics. These weights are then used to create weighted ensemble means and uncertainty estimates for the weighted ensemble mean. Given the key role of climate change prediction to effective climate change mitigation, adaptation and reduction activities, studies such as these aimed at reducing climate change uncertainty are vitally important.

A major and compelling reason for widespread confidence in any prediction or theory is the accuracy with which it has been able to predict events in the past. As such, historical observations are a natural data set to try and tune the weights associated with any ensemble weighting scheme.

To indicate the possibility that the authors proposed weighting approach would lead to more accurate climate change projections if tuned against historical observations, the authors remove one of the climate model simulations from their ensemble and treat it as if it were the unknown but historically observed truth. With this approach they can generate pseudo-historical observations and then optimize their weighting approach for that historical pseudo-data. They then checked whether the weights derived in this manner led to more accurate forecasts of the truth-model's state from 2080-2100. Fig. 4a shows the improvement over the sample mean as a function of their key tuning parameter for this historical data. This figure indicates that the optimal parameter value for the combined metric is between 0.3 and 0.5 (even though the text just gives it at 0.5). (The authors need to explain how they got 0.5 from Fig 4a rather than 0.4 or 0.3). Fig. 4b shows how the tuning parameter actually affects the 2080-2100 forecast accuracy, not for the combined metric but just two of the variables within the metric. Comparing Figure 4b with 4a shows that if one had used a good value of the parameter for the combined metric from the historical data, 0.4, say, the weighted multi-model mean would actually give a similar or less accurate precipitation and temperature forecast than the simple sample mean. This inability of the weighting method to produce significant forecast improvements when tuned against historical observations suggests

that the proposed method may be of little value. Nevertheless, there is merit in other aspects of the paper and with major revision; the paper could make a useful contribution to the field.

Specific Comments

The poor climate projection results obtained from the authors' proposed method when tuning using pseudo historical observations are in contrast to the findings of work I have been involved in. Specifically, in similar tests to those of Sanderson et al, Abramowitz and Bishop (2015, J. Clim) (AB) obtained average reductions in the root mean square distance from the out-of-sample truth greater than 30% when using the climate ensemble member weighting method of Bishop and Abramowitz (2013, Climate Dynamics) (BA). The current version of the paper lacks any reference to AB. Furthermore, on lines 67-68, it dismisses BA's approach as being undesirable for their North American application. This is incorrect. Small root mean square forecast errors is universally accepted as a desirable aspect of a forecasting scheme. AB showed that relative to the root mean square error of the uniformly weighted ensemble mean, the reduction in root mean square forecast errors due to the BA weighting method is profound. Furthermore, their method can easily be "geographically focused" for regions such as North America. As such, I strongly encourage the authors to revise their draft so that it acknowledges BA's approach as potentially useful for North America and discusses the positive results of AB.

Obviously, AB considered differing metrics to Sanderson et al. so no apples-to-apples comparison can be made between AB's results and the results of this paper but AB's work needs to be recognized and not dismissed as undesirable because of the BA method's use of metamodels. Each of BA's metamodels is a linear combination of the original models constructed so that the weighted mean formally minimizes error variance; and the BA ensemble variance is equal to this minimal value of the error variance. One needs to recognize that each raw climate model is itself a "meta Earth system" that is a crude approximation to the real Earth system. It is true that Bishop

and Abramowitz's metamodels are unrealistic in that, for example, they do not obey conservation laws for energy and mass. However, they are more realistic than the original models in the sense that their statistical relationship to historical observations is more like that of an ensemble of perfect models (replicate Earths) than the original models.

Having found rather poor forecasting results when using weights derived from pseudo-historical data, Sanderson et al. then consider weights that are tuned for model forecast data so that, on average, they deliver a weighted mean that is as close as possible to the 2080-2100 state of a climate model excluded from the set of ensemble members used for the forecast (Fig 4b). In statistics, such "in-sample" statistical tests are viewed with suspicion because of the possibility of overfitting. An additional concern about this approach is that it would be impossible to apply it to real observations (unless one waited until 2100 when the data would be available). One is left having to justify the approach on the assumption that the climate models are producing realistic future climate data. In contrast, if as in AB and BA, one demonstrated improved forecasts using historical observations, there would be much less room for argument about the realism of the data available for tuning. The revised paper needs to clearly address these concerns.

In addition to the aforementioned issue, the point by point comments below highlight other major and minor issues that, if addressed, would improve the paper.

Point by point and technical comments

1.Line 67-68. See above comments.

2. Sentence from line 74-76. Suppose that one had two simulations from a perfect model and that each was started with a different initial condition. In this case, the model for each of the simulations is the same even though, because of the chaotic nature of the Earth-system, the state estimates obtained will have differences. It can be shown that the mean of these two random perfectly realistic states would have considerably less distance from another perfectly realistic state (Bishop and Abramowitz, 2015). Hence, not including the second ensemble member simply because the model that produced it was identical to the model used for the first model would reduce the utility of the ensemble. Thus, this idea and its incorporation into the weighting scheme does not seem to be well justified. Perhaps the authors assumed that over a long enough averaging period the time-means of the two simulations would be identical. Long range modelling studies of low-frequency variability such as that of James and James (1989, Nature 342, 53 – 55) do not support this assumption. The revised paper should comment on this issue. 3. Section 3. Please add more details about the length and temporal filtering of the data set used to create the distance matrix.

4. Line 91-92 and Table 1. Extreme values such as "coldest day" are highly prone to large variations that are simply due to random sampling rather than any error in the distribution being sampled. One can easily prove this to oneself by sampling a normal distribution of 20x365 random normal numbers and seeing how much the minimum value changes. I did 12 such trials and found values ranging from -3.29 to -4.25. In contrast, if I look at the variation of standard deviations for 12 such trials I get values with the very small range of 0.98 to 1.01 – only 2% variation. By rewarding with high weights ensemble members that happen, by pure chance, to get extrema correct, you may be compromising the potential performance of your ensemble weighting technique. Why not use a standard deviation metric instead?

5. Caption of Fig 3. What does NCA4 stand for?

6. Subsection 3.5. It seemed that you held the independence weights constant for section 3.5. Please be clearer about how these were combined with the skill weights for the experiments reported on in Subsection 3.5.

7. Legend of Figure 4a. Are the "ta" and "tas" mentioned in this legend respectively the same as the "T" and "TS" mentioned in Table 1? The revised paper needs to ensure that Table 1 is consistent with this legend and vice-versa. Also, on my copy of the paper,

in Fig 4a it was extremely difficult to tell which line corresponded to which variable. It would be clearer if, in addition to color, you used shapes (triangles, boxes, diamonds, asterisks, etc) to help distinguish which line belongs to which variable.

8. Line 165. Here you state that Figure 4a suggests to you that 50% (0.5) minimizes forecast error. To my eye it looks like 0.4 or 0.3 minimizes forecast error. Please give more details about how you came up with the 50% value.

9. Line 191. Change "averages" to "averaged"

10. Line 198. Please provide more information about how you "skill weighted the ensemble". Does this create a new ensemble? How do you assess whether the truth lies within or outside of this skill weighted ensemble? I am unable to comment on any aspect pertaining to Fig 4c because of my uncertainty about what you actually did.

11. Weight normalization. The text is somewhat unclear about where and when the weights are normalized so that they sum to 1. Please be clearer about this. An equation stating exactly what you did would be helpful.

12. Figure 5. I like the idea of excluding similar models for the "model as truth" experiments. This option was not investigated by AB. Do your results change much if you don't exclude any models?

13. Line 216 – 218. State quantitatively what values are used. The previous sections used a whole range of values so it is unclear what precise values were finally chosen.

14. Line 430: Change "not trivial matter" to "not a trivial matter"

---

## Author Comment (AC1) · 4 Apr 2017

Thanks to the reviewer for their useful comments. We address each of the reviewer's points below, and attach a revised version of the manuscript addressing the concerns:

*The main limitation of the manuscript in my view is the primarily heuristic nature of the weighting schemes, which are at best partially justified. The introduction l73-78 sets out "two fundamental characteristics" of the scheme which are probably uncontroversial but which are not sufficient to narrow down the nature of the weighting scheme very much.*

We agree that our weighting scheme is heuristic, but we also think that it could be potentially useful. Clearly, one could conceive of other weighting schemes which satisfy the desired characteristics laid out in the introduction, and we do not suggest that our proposed approach is the only possible or the best solution. We simply propose it as

a strategy, and would welcome other contributions from the community with alternative strategies which allowed for a simultaneous consideration of model skill and replication. Due to the lack of direct verification of climate projections, it is fundamentally impossible to decide what method or model is best, and choices in any such method are necessarily subjective to some extent. Different choices will also work better or worse for certain applications. We argue what is needed is not a justification of a method being correct or best, but traceability of what the choices were, and how they could impact the results.

*I would however suggest that "relatively poor" would be more precise than the stated "demonstrably poor".*

Changed as suggested

*Taking performance weighting first, there is a substantial literature on this, albeit perhaps with limited results. Methods based on Bayesian Model Averaging (e.g. Hoeting et al 1999) have perhaps the strongest theoretical justification, but other approaches have also been presented (such as the "reliability ensemble averaging" approach of Giorgi and Means 2002). Olson et al 2016(a,b) present some recent applications of BMA to regional projections which seem highly relevant. I would ask the authors to consider whether their performance weights can be considered as Bayesian likelihoods, that is to say, is there an underlying statistical model which would result in this weighting scheme? If not, would it be worth changing to a more transparently presented and explained model, perhaps one which has been more widely applied and tested?*

We have added a section discussing BMA methods, and the REA method in the introduction. Notably, these methods are skill weights and do not easily allow for non-independent models. In BMA methods, a model's projection is weighted by its posterior model probability, which is largely independent of other models in the archive (apart from in the weak sense that the probabilities in the archive as a whole are normalized). So - the technique doesn't satisfy one of our two requirements. This is true of REA as well - but REA also carries the rather unjustifiable assumption that a model which produces a projection which is an outlier from the rest of the ensemble should be downweighted, which would arguably increase the model interdependency issue rather than address it. REA also leads to overly narrow uncertainties in the presence of many models (Knutti et al. 2010 J. Climate).

We've added the following on the topic of interpretation of the scheme: "It should be noted that although our likelihood weighting function is empirical, the functional form satisfies in a simple way the required parameters of the weighting scheme. The structure of this functional form is not fundamental, it can simply be shown to have some desired features. The technique is presented in this paper in a form which maximises clarity and reproducibility, but its effect can be described in Bayesian language. The total model weight is the posterior likelihood of a given model representing truth. Each model's prior probability of representing truth is given by its independence weighting, and the likelihood function is defined for the multivariate dataset using an assumed Gaussian likelihood profile in a space defined by the the sum of the normalized RMSE differences over all variables between each model and the observations."

*Of course any statistical method will necessarily rest on a number of assumptions and simplifications which may not be easily justified, but at least these could be presented explicitly. For example, while the distance factor Dq is considered as a tunable factor here, there is also the use of an exponential function which defines the weights, for which no explanation is given. Even without changing the overall structure of the weighting function, increasing the exponent from its value of 2 would result in a sharper cliff-edge at which weights drop from 1 to 0, and alternatively a lower exponent would result in a much more gradual change with weights more similar across the models. Is there a particular reason for the choices made here?*

We've tried to make it more clear in this version that the scheme is not intended to be *the* answer to weighting models. Yes, the functional form imposes some structural

limits on the weights one would obtain. By using a different power exponent, one could create a more or less polarized distinction between 'good' and 'bad' models - we could sample this dimension as another sensitivity study, but as you suggest, one could propose an infinite number of potential weighting functions, and we simply propose one which has some desirable characteristics, and we sample some useful parameters to sample a range of behavior - we claim no deeper interpretation than that. Given that Dq is chosen such that the method produces reliable uncertainties in the perfect model test, it is likely that a different exponent would lead to a different Dq but the overall mean and uncertainty would not change substantially.

However, there is precedent for using a Gaussian formulation for a likelihood function, we do not argue that our weighting scheme is not heuristic - our only requirement was to have a smooth, well behaved function which allocates maximum weight to a distance of zero, and no weight to a distance of infinity, without differentiating between two models which have distances « Dq. This actually leaves a rather limited set of choices for an appropriate functional form, for which a Gaussian structure is the simplest.

*Now moving on to the question of model independence, which here seems to be used to mean model output difference (as measured by a metric on output fields). The functional choice for the weighting again seems rather arbitrary. Since the goal of the parameter tuning seems to be to match the authors' beliefs that various models are replicated a particular numbers of times, is there a reason to use a function - which can only provide an approximation to this prior belief - rather than just use the authors' own judgements instead? For example a weight of 1/4 say could be applied to the GISS models directly, rather than trying to obtain a value close to this by tuning a single parameter. The choice of a fitted function seems to provide only a very thin veneer of objectivity to this subjective choice.*

Our argument for the representation of model interdependence is exactly that prior judgements of model interdependence are not required, because they are not always known - and this may be increasingly true in the future. As the reviewer points out, if the

only problem was to downweight models from the same institution which are known to be similar, the problem would be simple - either giving each of these models a fractional weight, or by taking only one version of institution's model.

However, in some cases, there are model interdependencies which cross institutions (take NorESM and CESM, or ACCESS and HadGEM). Unless the researcher knows about these in advance - they would miss them, whereas our method is data-driven, and if inter-depedencies are evident from the data, they are de facto considered. Interdependence will also vary on the quantity considered, two models may show similar behaviour in sea ice if they share the sea ice model, but differ more in other parts where components are not shared, or where other uncertainties dominate. We demonstrate our selection of the independence parameter using known cases, because in these cases - we know approximately what the answer should be. The point is then that the method can be generalised to cases where we don't know a priori the degree to which two models are related.

The constraints of this application are such that we were obliged to produce a single set of weights - but for the methodology in general, it allows for models to be assessed for interdependency conditional on certain outputs of the model which are relevant to the question in hand.

*C2 Despite these comments, I have no particular beef with the framework that has been presented - it does not look wrong or silly in any obvious way - but I also don't feel like I have been given any particular reason for using it. As outlined above, several of the numerous choices made don't appear to be that well justified. The tuning parameters do appear to have been selected sensibly, but this is only the last step after the creation of a structure that doesn't seem well supported.*

We hope that the above arguments help justify our approach, we propose a structure which a) satisfies our original requirements (downweight replication, upweight skill) in a framework which b) allows sufficient free parameters to tune for increased skill

without risking an overly calibrated result which might increase the risk of the truth lying outside the weighted ensemble distribution, and c) produces a single sets of weights for each model to be used in climate impact assessments based on a method easy to understand and implement by non-statisticians. Note that this paper is written to address a narrowly defined set of boundary conditions required by the author team of the Climate Science Special Report - specifically for a single set of weights which could be readily applied to a wide variety of projections. The method is not presented as fundamental, rather it is presented as a model which is defensibly fit for this particular purpose of dealing with a multi model ensemble in a National Climate assessment..

*A number of typos: 273-4 We briefly consider how the sensitivities of the method to different choices.* Corrected, thanks.

*322 taylor/tailor* Corrected.

*Fig 4 caption "1.5th percentile" really?*

Sorry -this was a version mixup. Now reworded to be consistent with the definition of $D_u$ in Figure 3.

Please also note the supplement to this comment:
http://www.geosci-model-dev-discuss.net/gmd-2016-285/gmd-2016-285-AC1-supplement.pdf

---

## Author Comment (AC2) · 4 Apr 2017

Thanks to the reviewer for his thoughtful reading and suggestions. We lay out below our thoughts in regard to his review, and how our paper relates to the author's work on the topic. We attach a revised version to address the reviewer's concerns and to better represent his own work on the topic.

*Fig. 4a shows the improvement over the sample mean as a function of their key tuning parameter for this historical data. This figure indicates that the optimal parameter value for the combined metric is between 0.3 and 0.5 (even though the text just gives it at 0.5). (The authors need to explain how they got 0.5 from Fig 4a rather than 0.4 or 0.3).*

The historical RMSE score isn't the only consideration, i.e. we don't only use Fig. 4a in our selection of the parameter - the value chosen is 0.8 or 80 per cent of the best-

model/obs distance. We did state that the lowest in-sample score was achieved with a value of approximately 0.5, but the next paragraph notes that this isn't how we choose our metric because choosing based on in-sample data only would lead to an overly confident constraint. Sorry for this confusion, we've reworded the first paragraph to make this clearer. We agree that the curve minimum in 4a is closer to 0.4 and have updated the text to reflect this, but note this was just an observation, this value is not used in any further analysis.

The two other factors considered are the out of sample (2080-2100) skill in Fig. 4b and the risk that our weighting would produce a distribution which increased the risk of the true model falling outside the weighted distribution. Hence - if historical RMSE was the only concern, we would choose a value of 0.4 - which would give us a better in-sample RMSE. The value of 0.8 is chosen such that the risk of overfitting is minimized, while still allowing for some moderate increase in weighted in-sample RMSE score.

*Fig. 4b shows how the tuning parameter actually affects the 2080-2100 forecast accuracy, not for the combined metric but just two of the variables within the metric. Comparing Figure 4b with 4a shows that if one had used a good value of the parameter for the combined metric from the historical data, 0.4, say, the weighted multi-model mean would actually give a similar or less accurate precipitation and temperature forecast than the simple sample mean. This inability of the weighting method to produce significant forecast improvements when tuned against historical observations suggests that the proposed method may be of little value.*

As noted above - overfitting the historical RMSE would reduce the out of sample skill, and we specifically don't do that for that reason. Hence, a less aggressive weighting was used - informed by Figs. 4b and 4c. Using the final value of 0.8, there is a small increase in out of sample skill - but we agree, it's a minor increase. But, we also don't find this particularly surprising - if there existed strong relationships between the mean state and the future temperature or precipitation changes, these would be exploitable emergent constraints in their own right. The literature has demonstrated

consistently that these constraints are rarely found in the CMIP archive. The fact that CMIP5 models on average agree better with observations than CMIP3 has not resulted in a more narrow projection range.

Our defense of the technique is that it provides a simple way to downweight clear model duplication, and relatively poor models in the archive. This may or may not result in a more accurate ensemble predictions, but there is no way to know whether a biased ensemble provides a biased projection that to see whether the weighting makes a difference. As we note, the actual CMIP archive has a tendency to have more replicates of models which exhibit lower RMSEs, there aren't many examples of models which exhibit huge biases both in the present, and there are no clear emergent constraints on future change - so the effect of the technique on CMIP5 is subtle because the model average happens to be almost optimal.

Our argument is that our method allows an analysis future-proofed against future ensembles with very poor models or with large numbers of replicates. If a group submits 1000 versions of the same model to CMIP6, our method would do a defensible job of allocating an appropriate amount of weight without modification. Similarly, if someone submitted a perturbed physics ensemble containing some model versions which were completely unlike Earth in the present, the presented method would downweight them appropriately. We agree that the use of the AB15 method has relevant merits for model weighting, but our needs in this case were specific, following a request for one set of positive model weights which could be used for further analysis by the authors of the report to address model skill and interdependence in a simple way.

*Nevertheless, there is merit in other aspects of the paper and with major revision; the paper could make a useful contribution to the field.*

**Specific Comments** *The poor climate projection results obtained from the authors' proposed method when tuning using pseudo historical observations are in contrast to the findings of work I have been involved in. Specifically, in similar tests to those of*

*Sanderson et al, Abramowitz and Bishop (2015, J. Clim) (AB) obtained average reductions in the root mean square distance from the out-of-sample truth greater than 30 percent when using the climate ensemble member weighting method of Bishop and Abramowitz (2013, Climate Dynamics) (BA). The current version of the paper lacks any reference to AB. Furthermore, on lines 67-68, it dismisses BA's approach as being undesirable for their North American application. This is incorrect. Small root mean square forecast errors is universally accepted as a desirable aspect of a forecasting scheme. AB showed that relative to the root mean square error of the uniformly weighted ensemble mean, the reduction in root mean square forecast errors due to the BA weighting method is profound. Furthermore, their method can easily be "geographically focused" for regions such as North America. As such, I strongly encourage the authors to revise their draft so that it acknowledges BA's approach as potentially useful for North America and discusses the positive results of AB. Obviously, AB considered differing metrics to Sanderson et al. so no apples-to-apples comparison can be made between AB's results and the results of this paper but AB's work needs to be recognized and not dismissed as undesirable because of the BA method's use of metamodels. Each of BA's metamodels is a linear combination of the original models constructed so that the weighted mean formally minimizes error variance; and the BA ensemble variance is equal to this minimal value of the error variance. One needs to recognize that each raw climate model is itself a "meta Earth system" that is a crude approximation to the real Earth system.*

We now devote a number of paragraphs to the description of the reviewer's 2013 and 2015 papers. AB15 is an interesting and novel framework for ensemble analysis, but it could never have been an option for this particular application because the request for the National Climate Assessment was specifically for one set of model weights which reflected model skill and independence. The weights were then passed to the author team, who conducted individual analyses for the NCA. As such, we were structurally constrained to produce a product which could be simply used by the author teams. A single set of weights could be incorporated fairly simply into the large number of predefined analyses which go into such a report (which is for general public consumption), whereas a transformation into statistical meta-models which do not, in themselves, follow physical laws would have been practically impossible to implement by the author team.

But - we do note that the comparison of 30 per cent reduction in out of sample truth is not comparing like with like. Firstly, the 30 per cent out of sample skill increase referred to in AB15 is the absolute difference between the mean state of the 'perfect' model and the optimized ensemble regression prediction in a period out of the training period. The out of sample skill in 4b in this paper is the skill in predicting the anomaly between present day T/P and the future. Part of the skill in AB15 comes from persistence of mean state bias - which is taken out of our test.

Secondly, although AB15 goes to some efforts to remove duplicates in their perfect model tests - they are not extensive. For example, AB15's "independent" test ensemble contains both CESM1 and NorESM1, and HadGEM2 and ACCESS - which both contain near replications of the atmospheric models. In this study, we have gone to significant efforts to remove any duplicates from our perfect model test, which would have trivially increased our out of sample skill.

*It is true that AB15's metamodels are unrealistic in that, for example, they do not obey conservation laws for energy and mass. However, they are more realistic than the original models in the sense that their statistical relationship to historical observations is more like that of an ensemble of perfect models (replicate Earths) than the original models.*

We would argue AB15's historical RMSE score is smaller by construction (there is no linear combination of models which could have a smaller RMSE), and the future reduction in anomaly projection error is not shown in AB15. But given that it is not empirically clear that one model subtracted from another is a physically meaningful quantity, only future anomaly error reduction in a true perfect model test where no close relatives of

the perfect model exist in the archive would constitute definitive evidence of greater skill. It could be argued that the any average of several models is also not necessarily physically meaningful because any combination of models no longer follows conservation laws, but a weighted average of models has a simple interpretation: a combined measurement of a number of models, weighted by their trustworthiness. Formulating the problem as a regression equation allowing negative coefficients though creates a more difficult product to interpret.

Given more models than degrees of freedom in the CMIP5 dataset, one could produce a near-perfect reproduction of the observations. Hence in order to be sure that AB15 is not subject to overfitting, it would be necessary to demonstrate that the degrees of freedom in CMIP models significantly exceed the number of fitted points. For a simple spatial field like temperature - where a few spatial modes can well define the response patterns of different models in the archive, this may not necessarily be the case.

*Having found rather poor forecasting results when using weights derived from pseudohistorical data, Sanderson et al. then consider weights that are tuned for model forecast data so that, on average, they deliver a weighted mean that is as close as possible to the 2080-2100 state of a climate model excluded from the set of ensemble members used for the forecast (Fig 4b). In statistics, such "in-sample" statistical tests are viewed with suspicion because of the possibility of overfitting.*

In our study (in contrast to AB15), we have only one parameter - so we don't have the ability to overfit in the regression sense of the word. We are not fitting to the future data directly, we are just reducing the degree to which the present day values can constrain the data if in the perfect model weighted average prediction of future anomalies can be demonstrated to be overconfident. Fig 4b is thus a diagnostic to show that if we had chosen an optimal value of the skill radius to maximise in-sample skill, then this would be non-optimal for out of sample skill. But the metric itself used to determine the parameters only considers historical data.

*An additional concern about this approach is that it would be impossible to apply it to real observations (unless one waited until 2100 when the data would be available). One is left having to justify the approach on the assumption that the climate models are producing realistic future climate data. In contrast, if as in AB and BA, one demonstrated improved forecasts using historical observations, there would be much less room for argument about the realism of the data available for tuning.*

We do apply the approach to observations - the constraints are entirely based on historical observations. We only use the future data in the models to assess how strong the costraints on past performance should be in general. A regression-based approach such as AB2015 has the capacity for overfitting, if the number of degrees of freedom exceed the number of models. Our technique calibrates a single parameter - which represents the degree to which historical data should weight a given model's future projection. The 2100 skill is a diagnostic, not a component of the weight and the method cannot 'fit' the combined model result to the 2100 data. Figure 4b simply says "if we over-constrain the models to their present day performance, then our prediction of future anomalies becomes less accurate". Therefore, we don't need the 2100 data from the real world to be able to use our method - we only use historical data - but 4b tells us that we should weaken that constraint from what we would have inferred from past performance alone. So we would argue that 4b is the opposite of overfitting, it explicitly weakens our constraint to to ensure against overfitting.

*The revised paper needs to clearly address these concerns. In addition to the aforementioned issue, the point by point comments below highlight other major and minor issues that, if addressed, would improve the paper.*

**Point by point and technical comments**

*1.Line 67-68. See above comments.*

We have significantly expanded this discussion in the light of the reviewer's comments.

*2. Sentence from line 74-76. Suppose that one had two simulations from a perfect model and that each was started with a different initial condition. In this case, the model for each of the simulations is the same even though, because of the chaotic nature of the Earth-system, the state estimates obtained will have differences. It can be shown that the mean of these two random perfectly realistic states would have considerably less distance from another perfectly realistic state (Bishop and Abramowitz, 2015). Hence, not including the second ensemble member simply because the model that produced it was identical to the model used for the first model would reduce the utility of the ensemble. Thus, this idea and its incorporation into the weighting scheme does not seem to be well justified. Perhaps the authors assumed that over a long enough averaging period the time-means of the two simulations would be identical. Long range modelling studies of low-frequency variability such as that of James and James (1989, Nature 342, 53 – 55) do not support this assumption. The revised paper should comment on this issue.*

This point is well taken, but it does not address the key aspect of the CMIP5 ensemble which we are trying to address - that all of the models are not perfect, and that some of them are near replicates of each other. Our technique does not throw out any models - but it allocates approximately equal fractional weights to near-identical models.

The relevant thought experiment is the following. Let's assume we have 3 models, 2 of these are structurally identical to each other, and the third has a different structure. Both of the structurally identical models have some underlying bias in their climate attractor, and the third model has a different bias - but the bulk errors are comparable.

In this case, knowing the above information - we would argue that the correct distribution of weight is $\frac{1}{4}$ for each of the structurally identical models and $\frac{1}{2}$ for the unique model, and this is the solution solved for in this paper. This conclusion has nothing to do with averaging periods (although clearly, the shorter the time series, the noisier the result will be).

Our previous work (Sanderson et al (2015b)) shows that the inter-model distances due to internal variability are an order of magnitude smaller than the differences between structurally dissimilar models in the CMIP archive, when evaluated using a similar metric to that used in this paper using 30 year climatological means. As such, the effect of bias due to model replication is well resolved in the context of noise generated by internal variability.

*3. Section 3. Please add more details about the length and temporal filtering of the data set used to create the distance matrix.*

We added the following paragraph: " Data from each model is taken from the first available initial condition member of each model's historical contribution to CMIP5. Data from years 1976-2005 are used from each model, averaging all years to form a monthly climatology. Data from the observations are monthly climatologies averaged from all available years within the 1976-2005 window."

*4. Line 91-92 and Table 1. Extreme values such as 'coldest day' are highly prone to large variations that are simply due to random sampling rather than any error in the distribution being sampled. One can easily prove this to oneself by sampling a normal distribution of 20x365 random normal numbers and seeing how much the minimum value changes. I did 12 such trials and found values ranging from -3.29 to -4.25. In contrast, if I look at the variation of standard deviations for 12 such trials I get values with the very small range of 0.98 to 1.01 – only 2 percent variation. By rewarding with high weights ensemble members that happen, by pure chance, to get extrema correct, you may be compromising the potential performance of your ensemble weighting technique. Why not use a standard deviation metric instead?*

Using a standard deviation assumes a normal distribution which is inappropriate for assessing the properties of the tail of the distribution. It also assumes that the distribution is bounded - and climate variables are not. The CSSR/NCA requires an assessment of extreme model behavior, and we use metrics from a well-established community

to form the statistics (we use the methodology laid out in Sillmann et al (2013), which shows such statistics are well sampled for a 20 year climatology - and we use 30). Note also that data at high temporal resolution is not always publicly available, whereas the standardized extreme indices are readily available for models and observations.

*5. Caption of Fig 3. What does NCA4 stand for?*

Expanded to the full name of the report.

*6. Subsection 3.5. It seemed that you held the independence weights constant for section 3.5. Please be clearer about how these were combined with the skill weights for the experiments reported on in Subsection 3.5.*

Text added to the paragraph: "In Figure 4(a), we use the uniqueness parameter Du determined in section 3.4 and sample a range of Dq."

*7. Legend of Figure 4a. Are the "ta" and "tas" mentioned in this legend respectively the same as the "T" and "TS" mentioned in Table 1?* The revised paper needs to ensure that Table 1 is consistent with this legend and vice-versa. Table 1 is now consistent in abbreviations.

*Also, on my copy of the paper, C5 in Fig 4a it was extremely difficult to tell which line corresponded to which variable. It would be clearer if, in addition to color, you used shapes (triangles, boxes, diamonds, asterisks, etc) to help distinguish which line belongs to which variable.* The figure has been reformatted for clarity as the reviewer suggests.

*8. Line 165. Here you state that Figure 4a suggests to you that 50 percent (0.5) minimizes forecast error. To my eye it looks like 0.4 or 0.3 minimizes forecast error. Please give more details about how you came up with the 50 percent value.*

We agree - we've changed the text. As explained above - this value was just an observation from the graph, it was not used in any part of the further analysis.

*9. Line 191. Change "averages" to "averaged"*

Done

*10. Line 198. Please provide more information about how you "skill weighted the ensemble". Does this create a new ensemble? How do you assess whether the truth lies within or outside of this skill weighted ensemble? I am unable to comment on any aspect pertaining to Fig 4c because of my uncertainty about what you actually did.*

We have considerably increased the length of this discussion.

*11. Weight normalization. The text is somewhat unclear about where and when the weights are normalized so that they sum to 1. Please be clearer about this. An equation stating exactly what you did would be helpful.*

Added equation 7.

*12. Figure 5. I like the idea of excluding similar models for the "model as truth" experiments. This option was not investigated by AB. Do your results change much if you don't exclude any models?*

Quite a lot, depending on the model and variable - not excluding clear replicates like NorESM/CESM tends to produce out-of-sample anomaly projection skill which is artificially high in the model as truth experiments. Keeping all members for the perfect model case therefore reduces the apparent out-of-sample skill a lot. Figure 1 (in this response) shows the equivalent of Figure 4b in the main document without prefiltering for near neighbours. The method would suggest a "model-as-truth" best average score of about 30 percent below the simple multi-model mean for precip, and 15 percent for temperature. I.e. It would give too much confidence in the out of sample skill.

*13. Line 216 – 218. State quantitatively what values are used. The previous sections used a whole range of values so it is unclear what precise values were finally chosen.*

Done.

*14. Line 430: Change "not trivial matter" to "not a trivial matter"*

Done

───────────────────────

[Figure]

Fig. 1.

[Figure]

**Supplement:**

[revised manuscript text omitted]

---

## Author Response (AR2)

Many thanks to both the editor and the reviewers for their careful reading of the manuscript. We attach an updated version addressing the points raised.

*49-51: "...ability of a model to project a certain change". It's not clear to me what this means. I assume you're talking about fitness-for-purpose here. E.g. a model's skill might vary for different regions of the world, or for different variables of interest. Please expand this sentence to make it clearer.*

Thanks for the comment, we have expanded the paragraph as follows:
"The weights should also be representative of the question at hand: skill is not a property of the model *per se*, but indicative of the ability of a model to project a certain change (Parker et al 2009). In other words, a climate model is fit for purpose if it can adequately represent the response of relevant physical processes in the required range of boundary conditions. This assessment of adequacy might change based on the regions and variables in question."

*126: "...contains on the ..." A typo, I think?*

Thanks. Fixed.

*139: should say "For each variable, v, ..."*

Fixed.

*144: "monthly climatology". But table 1 says "seasonal". Is there another aggregation step here? Or is one of these incorrect?*

Thanks for catching this. Sorry - version control. This version is seasonal data only.

*146-147: "area-weighted root mean square difference over the domain". This needs a little more explanation. Presumably, model output is gridded (but on different grids, depending on the model), and some (all?) of the observational data is gridded. I guess "area-weighted" suggests you've done something to handle the grid disparities. But I don't understand the steps involved in getting from the individual data points to an overall distance measure for each variable. Please elaborate (if only so that others could follow the same steps if they want to use the same analysis for other purposes).*

Added the following:
"All observations and model data are first linearly interpolated to a common 1 by 1 degree grid and 17 vertical levels. "

*147 "The matrix is then normalized". I think this should say "Each matrix...", or did I misunderstand? And is it normalized against the entire (inter-model) matrix mean, or just the mean distances of each model to the observations?*

Adjusted as follows:
"Each matrix corresponding to each variable is then normalized by the mean pairwise inter-model distance, such that for each field in Table 1, there is a n+1 by n+1 matrix representing the pairwise distance between each model (and the observations)."

*160: "The inter-model distance matrix is also computed from the inter-model distance matrix". This sentence doesn't make any sense.*

Sorry - yes. Corrected to:
"The independence weights can be computed from the inter-model distance matrix $\delta$. "

*202: The notation here is very awkward. Is it necessary to include the 20c superscript? You don't reference it anywhere in the text.*

It was for consistency with our previous paper where we used different time periods, but agreed - it makes little sense out of context here. Removed.

*Figure 6: Please label the x axis.*

Done

*386: "tailer" -> "tailor"*
Corrected